# Ticks and Tick-Borne Pathogens in Popular Recreational Areas in Tallinn, Estonia: The Underestimated Risk of Tick-Borne Diseases

**DOI:** 10.3390/microorganisms12091918

**Published:** 2024-09-20

**Authors:** Maria Vikentjeva, Julia Geller, Olga Bragina

**Affiliations:** 1Department of Communicable Diseases, Health Board, Paldiski mnt 81, 10614 Tallinn, Estonia; 2Influenza Centre, Health Board, Paldiski mnt 81, 10614 Tallinn, Estonia; 3Division of Chemistry, Department of Chemistry and Biotechnology, School of Science, Tallinn University of Technology, Akadeemia tee 15, 12618 Tallinn, Estonia

**Keywords:** ticks, *Borrelia*, urban, tick-borne pathogen, tick-borne encephalitis virus, *Borrelia miyamotoi*, *Neoehrlichia mikurensis*, *Anaplasma*, *Rickettsia*

## Abstract

This study reveals a significant presence of ticks and tick-borne pathogens in urban recreational areas of Tallinn, Estonia. During the period of May–June 2018, 815 *Ixodes* ticks were collected from an area of 11,200 m^2^ using the flagging method. Tick density reached up to 18.8 ticks per 100 m^2^, indicating a high concentration of ticks in these urban green spaces. Pathogen analysis demonstrated that 34% of the collected ticks were infected with at least one pathogen. Specifically, *Borrelia burgdorferi* s.l., the causative agent of Lyme borreliosis, was detected in 17.4% of the ticks; *Rickettsia* spp. was detected in 13.5%; *Neoehrlichia mikurensis* was detected in 5.5%; *Borrelia miyamotoi* was detected in 2.6%; and *Anaplasma phagocytophilum* and tick-borne encephalitis virus were detected in 0.5% each. These findings indicate that the prevalence and abundance of ticks and tick-borne pathogens in these urban environments are comparable to or even exceed those observed in natural endemic areas. Given the increasing incidence of Lyme borreliosis in Central and Northern Europe, the risk of tick bites and subsequent infection in urban recreational sites should not be underestimated. Public health measures, including enhanced awareness and precautionary information, are essential to mitigate the risk of tick-borne diseases in these urban settings.

## 1. Introduction

Ticks, especially those belonging to the *Ixodes ricinus* complex, are among the most medically important vectors for various diseases in the Northern Hemisphere. Traditionally the risk of a tick bite followed by Lyme borreliosis (LB) or tick-borne encephalitis (TBE) has been associated with deciduous or mixed boreal forests, pastures and meadows. However, with global urbanization and the development of green infrastructures and urban woodlands within the cities, various wildlife transformed from residing in wild landscapes to urban areas and adapted to new environments, becoming well established in close proximity to human settlements. This has increased the risk for various zoonotic and vector-borne diseases for members of the public.

Ticks are widely present in green spaces of urban environments, even despite low local floral and faunal diversity. Several studies from Europe report the detection of various tick-borne pathogens (TBPs) in ticks found within cities, such as tick borne encephalitis virus (TBEV), *Borrelia burgdorferi* sensu lato (BBSL), *Borrelia miyamotoi*, *Neoehrlichia mikurensis*, *Anaplasma phagocytophilum* and *Rickettsia helvetica* [1,2,3,4]. Many small mammalian species that are essential blood meal providers for larval and nymphal ticks and act as reservoirs for TBPs, such as BBSL and TBEV, are highly established in urban areas, well adapted to anthropogenic pressure and may thus play a role in the increasing incidence of tick-borne diseases (TBDs) in cities [5]. The urban heat island effect may also reflect positively on arthropods sensitive to cold temperatures by promoting their survival and shortening developmental diapause periods [6].

Estonia has long been an endemic area for TBE and LB, and by 2020, both diseases had reached record incidence rates. The incidence of TBE fell nearly threefold, reaching its lowest rate in the last 20 years at 5.1 cases per 100,000 people. By contrast, the number of LB cases reached a record high during the same period, with 182.1 cases per 100,000 people. This is three times higher compared to the levels recorded in 2013 [7]. Still, about 15% of diagnosed TBD patients with tick bites of an assumable geographical origin were reported within urban areas [8]. Two non-nidicolous *Ixodes* tick species of medical importance, *I. ricinus* and *I. persulcatus*, have been shown to maintain the circulation of various TBPs in Estonia. In addition to European (TBEV-Eu) and Siberian (TBEV-Sib) TBEV subtypes, at least five species of BBSL, as well as various TBPs including tick-borne Rickettsiales (*A. phagocytophilum*, *Ehrlichia muris*, *N. mikurensis*, *R. helvetica*, *R. monacensis* and *Candidatus R. tarasevichae*), relapsing fever group *B. miyamotoi* and several *Babesia* species have been detected in questing ticks in their natural habitats in Estonia [9,10,11,12,13,14,15]. The aim of this study is to investigate the distribution and abundance of ticks in urban areas of Tallinn and to analyze these ticks for the presence of tick-borne pathogens (TBPs), including TBEV, BBSL, *B. miyamotoi*, *A. phagocytophilum*, *N. mikurensis* and *Rickettsia* spp., in popular recreational and leisure sites within the city.

## 2. Materials and Methods

### 2.1. Study Areas Selection and Tick Collection Sites

Two main aspects were considered when choosing study areas: suitable habitats for ticks and popularity among visitors. Satellite imagery from Google Earth and the Estonian Land Board Web Map application (xgis.maaamet.ee/maps accessed on 15 April 2018), as well as personal observations and data on recreational areas from the official Tallinn webpage (www.tallinn.ee, accessed 15 April 2018), were used for the initial identification of potentially suitable study areas, which were then visited to determine specific flagging sites. Areas were selected according to the presence of bushes, a broad-leaf or temperate forested area and a litter layer, with known recreational popularity among visitors. Based on these criteria, 14 sites were selected in collaboration with the respective authorities where applicable (Figure 1). 

Each transect was located along trails or their closest proximity to imitate visitors’ behavior as closely as possible. The observations and variables recorded at each site included a date, time, transect length and air temperature. Habitat and vegetation type, as well as signs of any host presence by seeing an actual animal or any direct evidence (e.g., tracks, birdsongs, nests, feces), were also noted.

Based on climate and weather observations from previous years, the month of May was predicted to be the most suitable time for the survey. Sites were surveyed for ticks from 9 May to 1 June 2018, preferably during morning hours from 9 to 11 a.m. Each site was examined once.

### 2.2. Tick Collection and Species Identification

Tick collections were performed using the flagging technique with a 1 m^2^ light-colored flannel cotton cloth, attached to a wooden T-shaped handle, that was dragged over vegetation at slow pace over a set of distances of 5 m^2^. At each survey site, a minimum of 60 × 5 m^2^ transects were completed by one person. More transects were carried out at sites with less tick density (with no nymphs or adult ticks observed in first 30 transects, i.e., 150 m^2^) or in sites with uneven terrain. The minimum survey area for each site was 300 m^2^.

Ticks were removed from the cloth with tweezers, placed into separate glass vials according to their development stage and sex and stored at +4 °C prior to species identification. The presence of larvae was noted, but they were not collected, counted nor included in any analysis of this study. Adults and nymphal tick species were identified individually using a stereomicroscope according to morphological keys [16]. Ambiguous specimens were additionally identified by molecular keys using PCR based on internal transcribed spacer 2 (ITS2) and the partial 16S rRNA gene, as described previously [11,17].

Questing tick abundance was calculated as the density of total number of ticks (DOT), adults (DOA) and nymphs (DON) collected per 100 m^2^. Additionally, indexes of abundance, which takes into account collection efforts, were calculated as described [18]. The DIN per 100 m^2^ was derived for each site by multiplying the mean density of host-seeking nymphs (DON) by the infection prevalence at each site.

### 2.3. DNA/RNA Isolation

All ticks were individually processed for DNA and RNA isolation using the blackPREP Tick DNA/RNA kit (Analytik Jena, Jena, Germany). The initial tick lysis step was increased twice in time, and the homogenization step was performed twice according to manufacturers’ recommendations. Homogenization was performed using MixerMill MM301 (Retsch, Haan, Germany). Mixing mill cassettes with vials, containing tick homogenate, were flipped over between homogenization steps to assure better milling performance. DNA and RNA solutions were then stored at −20 °C and −70 °C, respectively, prior to further individual screening for the presence of TBPs.

### 2.4. TBP Detection

All tick nucleic acid extracts were analyzed individually. Positive tick samples, which were obtained from previous studies and successfully sequenced for respective TBP gene fragments, and deionized PCR-grade water were used as positive and negative controls, respectively, at each amplification step. To reduce contamination risk, each procedure, including tick pre-extraction washes, RNA/DNA extraction, PCR reaction mix preparation, DNA/RNA adding step, PCR reaction and gel-electrophoresis, were performed in separate rooms.

The initial screening of *Rickettsia* spp. and TBEV was performed by real-time PCR based on a 74 bp fragment of the *gltA* gene and 67 bp long fragment of a 3′ non-coding region, respectively [12,19]. The detection of other bacterial TBPs was performed by nested PCR reactions based on respective TBP specific regions as follows: 245–256 bp long 5S–23S intergenic spacer (IGS) region was used for BBSL [20], 532 bp fragment of the *p66* gene was used for *B. miyamotoi* [21] and 1350 bp 16S rRNA fragment was used for *Anaplasmataceae* [11]. Detailed information on all PCR reactions, including target name, product size, primer/probe sets and amplification conditions used in this current study are presented in Appendix A.

### 2.5. TBP Identification and Genotyping

After initial determination, all positive PCR samples were subjected to direct Sanger sequencing with subsequent BLAST analysis of obtained sequences for TBP species identification. Sequencing of PCR products was performed at the Core Facility, Institute of Genomics, University of Tartu. BLASTN^®^ tools (https://blast.ncbi.nlm.nih.gov/Blast.cgi, accessed 21 January 2021) were used for sequences, comparing them to existing GenBank records once MEGA 7.0 software had been used for sequencing chromatogram proofreading, sequence trimming and group aligning.

The genotyping of *Rickettsia* species in positive qPCR samples was performed by the sequencing of the 667 bp long region of the *gltA* gene, amplified by nested PCR [22]. Samples positive for *Rickettsia* spp. qPCR, but negative for nested PCR of the *gltA* fragment, were additionally subjected to PCR amplification and sequencing of a 769 bp long *ompB* gene region, as described by Roux and Raoult [23], and an 843 bp long region of the *sca4* gene as described by Igolkina et al. [24]. Samples remained negative for any analyzed gene region by PCR, but those with Ct values below 35.0 were classified as unspecified.

TBEV genotype detection for RT-qPCR positive samples with Ct values below 35.0 was performed via nested RT-PCR amplification of the partial E protein gene 465 bp long fragment [14,19].

All final nested PCR products were visualized by 1% agarose gel electrophoresis, stained with ethidium bromide.

Due to the number of identical sequences, especially within BBSL and *R. helvetica*, only unique sequences were deposited, and duplicate sequences were omitted from submission. Samples whose nucleotide sequences allowed for performing pathogen genotyping, but with a partially poor chromatogram or with possible mixed infections of several pathogen species strains, were also excluded from depositing. All other sequences were deposited with the maximum reliable plot length read. Nucleotide sequences of gene fragments obtained during this study were submitted to the GenBank database under the following accession numbers: MW924118 and MW924120—MW924135 for BBSL species (196-215 bp 5S-23S IGS), MW924974—MW924983 for *B. miyamotoi* (*p66*, 472 bp), MW924984—MW925040 for *R. helvetica* (554 bp *gltA*, 739 bp *ompB*, and 715 bp *sca4*), MW922757—MW922793 for *N. mikurensis* (618-118 bp 16S rRNA gene fragment), MW922752—MW922756 for *A. phagocytophilum* (1158 bp 16S rRNA gene fragment) and MW916612—MW916613 for TBEV (465–468 bp long E protein gene fragment).

### 2.6. Statistical Analysis

Confidence intervals with 95% (CI) for TBP prevalence were calculated using the Wilson score interval. A Chi square test was used to evaluate the significance of the results from each study site. Statistical analyses were performed using GraphPad Prism 9. Since the collection of ticks at each site was carried out once and the choice of site was not based on the landscape, habitat or vegetation preferences, no statistical analysis for tick distribution was performed for any study site. No habitat preferences, climate or any other variables among sites was applied for this study.

## 3. Results

### 3.1. Tick Sampling

A total of 177 adults (79 male and 98 female ticks) and 638 nymphs were collected from over 11,000 m^2^ of the vegetation screened (Table 1). All ticks were identified as *I. Ricinus*, except one *I. persulcatus* male collected in the Sütiste forested park (site 2). Among the 14 visited sites, the highest numbers of ticks were collected at Tallinn Zoo, Estonian Open Air Museum and Pirita forest park (sites 12, 8 and 2, respectively), which accounted for 28.7% (234/815), 27.7% (226/815) and 15.0% (122/815) of the total number of collected ticks, respectively. The estimated overall mean DOT was 9.3, with the greatest at Estonian Open Air Museum, Männiku forest (site 14) and Tallinn Zoo (18.8, 18.3 and 15.6, respectively) (Table 1). The overall mean density of nymphs (DON) was 7.3 nymphs per 100 m^2^, although it fluctuated between 1.3 and 8.6 among the sites (Table 1).

From all the sites visited during this study, ticks were found at 10 sites. There were no ticks found in the city central parks Hirvepark and Toompark; von Glehni park; Järve health trails; and Sanatooriumi park (sites 5; 6; 11; and 13; respectively) (Figure 1).

### 3.2. Pathogen Prevalence

Of all ticks analyzed, 34% tested positive for at least one TBP species. Between sites, the TBP abundance rate varied from 3.3% in the Nõmme-Mustamäe forested area to 43.8% in the Estonian Open Air Museum (Table 2), both with extreme statistically significant relations to other sites (χ^2^ = 13,045, df = 1, *p* = 0.0003 and χ^2^ = 13,434, df = 1, *p* = 0.0002, respectively). Other statistically significant TBP abundance scores were obtained in ticks collected from Tallinn Zoo (41.5%; χ^2^ = 8153, df = 1, *p* = 0.0043) and Männiku (19.2%; χ^2^ = 7831, df = 1, *p* = 0.0051) (Table 2).

### 3.3. Borrelia burgdorferi (Sensu Lato)

*B. burgdorferi* s.l. was the most frequently detected tick-borne pathogen (TBP), identified in ticks from nearly all collection sites except for the Pirita River Valley. Overall, BBSL DNA was detected in 17.4% of all analyzed ticks, representing 51.2% of all TBP-positive ticks. The prevalence of BBSL varied between sites, ranging from 3.3% at Nõmme-Mustamäe to 25.2% at the Estonian Open Air Museum (Table 2). Statistically significant differences in BBSL prevalence were observed among several sites, including Pirita River Valley, Tallinn Zoo and Männiku, with χ2 values of 4.298 (df = 1, *p* = 0.0382), 13.217 (df = 1, *p* = 0.0003), 3.884 (df = 1, *p* = 0.0488), 5.254 (df = 1, *p* = 0.0219) and 7.95 (df = 1, *p* = 0.0048), respectively.

A sequence analysis of the 5S–23S intergenic spacer (IGS) region revealed the presence of three BBSL genospecies. The predominant genospecies was *Borrelia afzelii* (121/142; 85.2%), followed by *Borrelia garinii* (11/142; 7.3%) and *Borrelia valaisiana* (7/142; 4.7%). Three samples could not be specified at the genospecies level due to inadequate sequencing results, which might indicate the presence of mixed genospecies (Table 2). The nucleotide similarity of the *B. afzelii* 5S–23S IGS sequences ranged from 77.4% to 99.5% among samples and matched previously detected sequences from Estonian questing and passerine-attached ticks (GenBank accession numbers KX418639, KX418638, KX418640), as well as sequences reported from France (KY273112, KY273113), Italy (MT038899), Slovakia (KX906933, KX906945), Taiwan (JX649207) and Russia (MK118750, AB178349). The nucleotide similarity rates of *B. garinii* 5S–23S IGS sequences varied from 79.0% to 99.5% and clustered with sequences previously reported from Estonia (KX418634 and KX418637), Taiwan (JX649205), Italy (MT038900), Belarus (AY772205), Sweden (JX909934), Czech Republic (AF497993) and Russia (MK118761). The *B. valaisiana* sequences were identical to each other and to those previously detected in Estonian *I. ricinus* from the common blackbird (KX418641), as well as to strains reported from Spain (MG245790), the Czech Republic (AF497989) and Italy (MT038902).

### 3.4. Borrelia miyamotoi

*B. miyamotoi* was detected in 2.6% of all analyzed ticks (21/815). This genospecies was found mostly in the Estonian Open Air Museum followed by Tallinn Zoo and Männiku forest (Table 2), whereas significant association between study site and the prevalence rate was obtained for the Open Air Museum only (χ^2^ = 4255, df = 1, *p* = 0.0391). Sequencing analysis of the *B. miyamotoi* partial *p66* gene showed that nucleotide sequences of this study are identical to each other and to sequences revealed previously in the Estonian tick population [10] and from human patient samples from Sweden (MK458691).

*Rickettsia* spp. was the second most prevalent bacterial TBP after BBSL. Its presence was detected in 13.5% (110/815) of all analyzed tick samples and varied from 3,2% to 22,2% between study sites (Table 2), whereas statistically significant prevalence was shown only in Nõmme-Mustamäe (χ^2^ = 4.86, df = 1, *p* = 0.0275). According to sequencing analysis of partial *gltA*, *sca4* and *ompB* gene nucleotide sequences, all *Rickettsia*-positive samples belonged to the *R. helvetica* species and were identical to each other within each gene fragment and to sequences reported from Western Siberia (KX963385), Germany (MF163040, KU310591) and to those previously detected in Estonian *Ixodes* ticks [12].

### 3.5. Anaplasmataceae

Although total *Anaplasmataceae* DNA prevalence rate shown was 6.1% (50/815), these TBPs were detected only in ticks collected from three sites: Tallinn Zoo, Estonian Open Air Museum and Pirita forest park (Table 2). Of these sites, only Tallinn Zoo showed a statistically important association between the collection site and obtained prevalence (χ^2^ = 14,115, df = 1, *p* = 0.0002). The analysis of *Anaplasmataceae* 16S rRNA sequences revealed the presence of two species: *A. phagocytophilum* (4/815; 0.5%) and *N. mikurensis* (45/815; 5.5%) (Table 2). One sample remained unspecified and was thus treated as *Anaplasmataceae* sp.-positive.

The most prevalent *Anaplasmataceae* species detected was *N. mikurensis*. *N. mikurensis* was identified in *I. ricinus* ticks collected from Pirita Forest Park (4 out of 122; 3.3%), the Estonian Open Air Museum (14 out of 226; 7.5%), and Tallinn Zoo (24 out of 234; 10.2%). Within partial 16S rRNA gene *N. mikurensis* sequences, this study showed 98.2–99.6% similarity to GenBank sequences reported previously from Estonian ticks (KU535862) and up to 99.4% similarity to sequences from Germany (KU865475) and Western Siberia (MN736126).

*A. phagocytophilum* was the least detected *Anaplasmacateae* species, detected in four *I. ricinus* ticks from Pirita forest park (2/122; 1.6%), Estonian Open Air Museum (1/226; 0.4%) and Tallinn Zoo (1/234; 0.4%). The 16S rRNA partial nucleotide sequences of *A. phagocytophilum* obtained in this study were 99.7–99.9% similar to each other. A comparison to previously reported sequences from Estonian questing ticks (acc. no HQ629920, HQ629922, HQ629920) and sequences reported from Russia (acc. no HQ629911), Sweden (acc. no AY527213) and Austria (acc. no JX173652) showed 99.7–100% similarity.

### 3.6. TBEV

According to qRT-PCR results, TBEV was the least common of the detected TBPs, detected in 4 *I. ricinus* nymphs of all 815 examined individual ticks (total prevalence of 0.5%) found at Pirita river valley (1/18; 5.6%), Ilmarise health trails (1/37; 2.7%), Estonian Open Air Museum (1/226; 0.4%) and Männiku forest (1/73; 1.4%) (Table 2).

Two samples were successfully sequenced and genotyped. According to the analysis of the partial E gene sequence obtained from an *I. ricinus* tick sample from the Estonian Open Air Museum, it clustered with TBEV-Sib sequences previously detected in Estonian *I. persulcatus* ticks collected in Eastern Estonia (acc. no KT748749 and KT748748) at an identity rate of 99.8%, belonging to the Baltic lineage within TBEV-Sib [14,22]. Another TBEV partial E gene sequence, retrieved from an *I. ricinus* sample collected at Ilmarise health trails, clustered within the TBEV-Eu subtype with 98.7% similarity to the strain previously reported from Estonia (acc. No GU183383) and 99.6% similarity to the TBEV strain from Latvia (acc. No AJ319583).

### 3.7. Mixed Infections

In all, 15.2% (42/277) of all TBP-positive ticks contained double infections, and four tick samples tested positive for three tick-borne pathogens (4/277; 1.4%). The most frequently detected TBP combination in double infected ticks was *B. afzelii* with *N. mikurensis* (18/42; 42.9%) or *R. helvetica* (13/42; 31.0%), and these originated mainly from Open Air Museum, Tallinn Zoo and Pirita forest park. It is noteworthy that of the four TBEV-positive tick samples, two were co-infected with bacterial TBP; one tick sample from the Estonian Open Air Museum was positive for TBEV-Sib and *N. mikurensis*, and the TBEV-Eu-positive sample from Männiku was also positive for unspecified BBSL. Of the four tick samples with triple infections, three tested positive for the presence of *B. afzelii*, *R. helvetica* and *N. mikurensis*, and one tested positive for *B. afzelii, R. helvetica* and *B. miyamotoi*.

## 4. Discussion

Prior to this study, there was no previous information on the spread of ticks and TBPs in urban recreational areas in Tallinn, Estonia. This study confirms the presence of *Ixodes* tick species in popular recreational, outdoor sport and leisure areas in the largest city of Estonia, with abundance rates that are comparable to or even exceed those previously recorded in the most endemic foci in natural environments. Notably, among all collected ticks from the genus *Ixodes*, we identified the species *Ixodes ricinus*, except for a single tick that belonged to the species *Ixodes persulcatus*. This finding underscores the diversity of tick species present in urban settings and highlights the potential public health risks associated with these areas [22]. As shown in this study, large and less-fragmented areas, such as Tallinn Zoo, Open Air Museum, Pirita forest park and Männiku, with needle- and broad-leaved trees, underwood with rich litter and signs of the presence of an ample variety of synanthropic small and medium-sized mammals offer a prerequisite environment for tick survival, development and maintenance. Therefore, a higher number of ticks, especially nymphs, is expected (DOT over 9.8 and DON over 8.6). By contrast, smaller, highly managed parks with mowed areas, such as Kadrioru park, Hirve/Toompark and von Glehni park, showed significantly lower tick densities (DOT < 2; DON < 1.3), despite having similar vegetation. While not focusing on tick density, our study results are generally in agreement with other European studies [23]. This suggests that human activity and landscape management practices may play a significant role in reducing tick habitat suitability in urban parks.

The prevalence of tick-borne pathogens varies significantly across European countries. The prevalence rate of 34% identified in this study is notably high compared to other regions, such as Spain, where the prevalence of tick-borne pathogens in urban environments was only 4.1%, or in French peri-urban forests, where it reached 15.9% [25,26]. The high average prevalence rate of TBPs in Tallinn’s urban green spaces indicates a significant risk of tick-borne infections, potentially even greater than in more extensive natural habitats due to the limited availability of green areas within the city [27].

Lyme borreliosis is notifiable and of great concern in Estonia. According to Estonian Health Board epidemiological reports from 2014–2018, up to 19% of all confirmed LB patients in Harju county with tick bites of known geographical origins had been bitten by ticks within Tallinn [8]. The prevalence rates of *B. burgdorferi* s.l in our study (17.4%) are compatible with those reported in Scandinavia, the Balkans and Central Europe (15.5%, 18.5% and 19.3%, respectively) [28,29]. Similar results have also been shown in urban areas in Switzerland (18%) and the urban Lazienki Park in Warsaw, Poland (17.3%) [23,30]. However, the BBSL prevalence reached 23.0% in Helsinki, Finland, demonstrating variability across different urban environments [31]. The presence of three LB-associated species—*B. afzelii*, *B. garinii*, *B. valaisiana*, known to be the most prevalent in Europe [32,33,34]—is in agreement with previous results concluded in Estonia [9]. As seen in previous studies, in Europe, *B. afzelii* and *B. garinii* were the most prominent species found in *I. ricinus* ticks. Notably, *B. afzelii* is predominant in Nordic countries (also in Mediterranean ones), whereas *B. garinii* and, to a lesser extent, *B. valaisiana* have been brought to Nordic countries (e.g., Sweden) by migratory birds, particularly pheasants [35] The predominance of *B. afzelii* and *B. garinii* in urban *I. ricinus* ticks (85.2% and 7.7%, respectively) [9], compared to natural environments (56.1% and 20.3%), is an interesting finding. A similar disproportion has also been observed in Poland [23], Switzerland [30] and Belgium [36]. This disparity may be explained by the dilution/amplification effects in fragmented urban landscapes and the availability of hosts. Natural forests and other sylvatic areas with little anthropogenic disturbance, fragmentation and transformation are inhabited or visited during migration stops by large amounts and varieties of avian species, which might serve as sources of avian-associated TBPs, such as *B. garinii* and *B. valaisiana* [32,37,38]. Thus, under conditions of anthropopressure, the contact of ticks with birds and the prevalence of bird-specialized TBPs are significantly decreased in comparison to natural areas, which may lead to a lower presence of these TBPs in urban ticks. By contrast, rodents, which are highly adapted to an urbanized environment, smaller agglomeration-surrounded areas and human interruption, promote sub-adult tick population maintenance, facilitate the frequency of tick–host contacts and trigger an increase and amplification of rodent-associated *Borrelia*, like *B. afzelii* [36]. 

*B. miyamotoi*, which belongs to the Hard Tick-Borne Relapsing Fever (HTBRF) group, has been documented to undergo vertical transmission in *Ixodes* ticks [39]. The detection rate of *B. miyamotoi* in our study was 2.6% across all analyzed *I. ricinus* ticks, with site-specific prevalence rates ranging from 0.75% to 4.4%. These findings are consistent with rates observed in urban and suburban areas in Slovakia [40], Poland [41], France [42] and Switzerland [30], as well as with our previous results from the Valgamaa and Võrumaa counties, where *I. ricinus* co-circulates with *I. persulcatus*, and the highest positivity rates for *B. miyamotoi* were recorded at 2.8% [10]. As the population density of peridomestic mice and voles might be higher in urban regions due to favorable breeding and survival factors [43] and as *B. miyamotoi*, being related to relapsing fever *Borrelia*, is transovarially transmitted, the higher infection rates in urban areas versus natural wooded sites may be due to higher amplification rates of this pathogen within urban landscape fragments compared to larger natural woodlands and pastures.

Our previous studies indicated the circulation of at least four medically important species within the *Rickettsiales* order in Estonian *I. ricinus* populations: *A. phagocytophilum* [12], *N. mikurensis* [11], *R. helvetica* and *R. monacensis* [13].

A spotted-fever *Rickettsia* group is transmitted trans-stadially and transovarially among ticks, and, with high tick abundance in urban green areas, may contribute to higher infection rates of this pathogen. The detection of different *Rickettsia* spp., such as as *R. monacensis*, *R. raoultii*, *R. slovaca* and *R. Helvetica,* in urban and suburban *I. ricinus* ticks has been reported from Germany, the Czech Republic, Poland, Ukraine, Romania and Slovakia at rates from 1 to 47% [44,45,46,47,48,49,50,51]. In this study, no other Rickettsia species than *R. helvetica* has been detected. Although the total abundance of Rickettsia is in line with that detected in natural tick habitats in Harjumaa county previously, larger green open spaces within the city showed up to twice as high rates of Rickettsia distribution (up to 19.5% vs. 10.2%, respectively) [13]. 

*N. mikurensis* appeared to be also slightly more abundant in urban *I. ricinus* ticks compared to those found in the natural *I. ricinus* allopatric areas with a site-specific prevalence of 3.3–10.9% (average 5.5%) versus 1.0–9.1% (average 0.9%), respectively [11]. These findings align with studies conducted in urban and sylvatic areas in Austria, Sweden, Germany and Poland, which reported prevalence rates from 4.2% to 19.3% [52,53,54,55]. Such widespread prevalence of the pathogen may be connected not only to arthropod vectors but also to reservoir hosts—bank voles and yellow-necked mice—which are largely synurbanized. Some studies also claim that non-rodent species such as hedgehogs, but not insectivores, may also contribute to *N. mikurensis* maintenance in urban and peri-urban green landscapes and human dwellings [56,57].

The detection of *A. phagocytophilum* at a prevalence rate of 0.5% makes it one of the least abundant pathogens found in this study, consistent with our previous research in natural landscapes, as well as with studies from Austria (1%) and Poland (0.8–1.2%) [12,53,58,59]. Also, the scattered distribution of this TBP, which in this study was concentrated mainly in larger, mostly outskirt city landscapes—the Pirita forested area, Zoo and Estonian Open Air Museum—corresponds to the presence of synurbanized ungulates or other known *Anaplasma* animal reservoir hosts, essential for its maintenance and circulation in nature. Although *A. phagocytophilum* was detected in sites with the highest tick densities as well, the absence of it in the least human-disturbed and fragmented Männiku forest may be due to a dilution effect. Thus, the distribution of *A. phagocytophilum* in urban conditions might be less dependent on ticks and small mammals present but rather on suitable habitats for ungulates.

The circulation of the tick-borne encephalitis virus (TBEV) in natural foci is maintained by small rodents, which serve as competent reservoir hosts, and ticks, which act as both hosts and vectors [60]. As many rodent species are well adapted to a human-affected and urbanized environment, the presence of TBEV foci and, therefore, the occurrence of autochthonous human TBE cases even within large cities is possible [61]. Previous studies have found TBEV in questing ticks with prevalence rates ranging from 0.2% to 0.8% in the *I. ricinus* allopatric area and up to 4.9% in areas where *I. persulcatus* also circulates [15]. This is consistent with the results of this study, as well as with TBEV prevalence rates in *I. ricinus* ticks in European foci [62]. According to epidemiological data of the Estonian Health Board, about 18% of TBE patients in Harju county had a tick bite history from Tallinn [8]. The results of this study not only confirm the presence of TBEV foci in green areas within the city but also indicate the circulation of European and Siberian subtypes of TBEV in *I. ricinus* ticks within urban habitats. Since the presence of TBEV-Sib in *I. ricinus* ticks in locations with no *I. persulcatus* co-circulation had also been previously shown [15], it may be assumed that TBEV-Sib might be potentially spread into *I. ricinus* distribution areas without the presence of its principal vector, *I. persulcatus*.

Due to the fragmentation and separation of urban green areas, ticks collected in these territories most likely have several pathogens at once/at the same time, since they most likely feed on the same hosts. In general, it has been argued more than once that for *Ixodes* ticks, co-infections are rather the rule than an unusual phenomenon. The most prevalent co-infection combinations were *B. afzelii* with *N. mikurensis* (42.9%), as well as *B. afzelii* with *R. helvetica* (31.0%); previously, the same situation was observed and described in Switzerland [63] due to their common reservoir. A study of urban ticks in Romania reported a co-infection rate of 34.3% among all *I. ricinus* ticks. The most common dual co-infections involved *Rickettsia* spp. and *Borrelia* spp., followed by co-infections with *Rickettsia* spp. and *A. phagocytophilum*, as well as *A. phagocytophilum* and *Borrelia* spp. [44]. The presence of multiple pathogens may facilitate more effective colonization of the host, driven by synergistic interactions initiated through co-transmission and the simultaneous invasion of the host by multiple pathogens. 

## 5. Conclusions

This study focuses on the city of Tallinn as a model for understanding tick populations and the prevalence of tick-borne pathogens (TBPs) in urban environments. It reveals a well-established presence of *Ixodes* ticks, particularly *Ixodes ricinus*, across both central and peripheral green spaces of the city. The overall tick density in Tallinn’s urban areas varied significantly, with higher concentrations observed in larger, less-fragmented green spaces such as Tallinn Zoo, the Estonian Open Air Museum, Pirita forest park, and Männiku. However, tick density was generally low in smaller, more intensively managed green areas. The prevalence of *Borrelia* spp. and other pathogens was relatively high, with 34% of ticks testing positive for at least one pathogen. The most commonly detected pathogen group was *B. burgdorferi* s.l., particularly in areas with ecological connectivity to the outskirts or with dense vegetation. Infected ticks were found in 26.7% of all samples, with the highest prevalence recorded at the Estonian Open Air Museum. Wildlife such as rodents, deer, boar and birds are likely key contributors to the presence of infected ticks in these urban settings, serving as hosts that maintain and spread tick populations. This study also identified several other pathogens, including *B. miyamotoi*, *R. helvetica* and *N. mikurensis*, suggesting a complex ecology of TBP transmission within the city. Given the increasing popularity of urban green spaces for recreation, public health awareness regarding tick-borne diseases is crucial. Our study highlights the need for targeted tick management and control strategies, especially in areas with high tick densities and pathogen prevalence. Urban planning should consider the ecological characteristics that support tick populations and pathogen transmission, potentially reducing the risk of tick-borne diseases in urban settings. Future research should focus on understanding the dynamics of TBP transmission and the role of urban wildlife hosts in sustaining these pathogens.

## Figures and Tables

**Figure 1 microorganisms-12-01918-f001:**
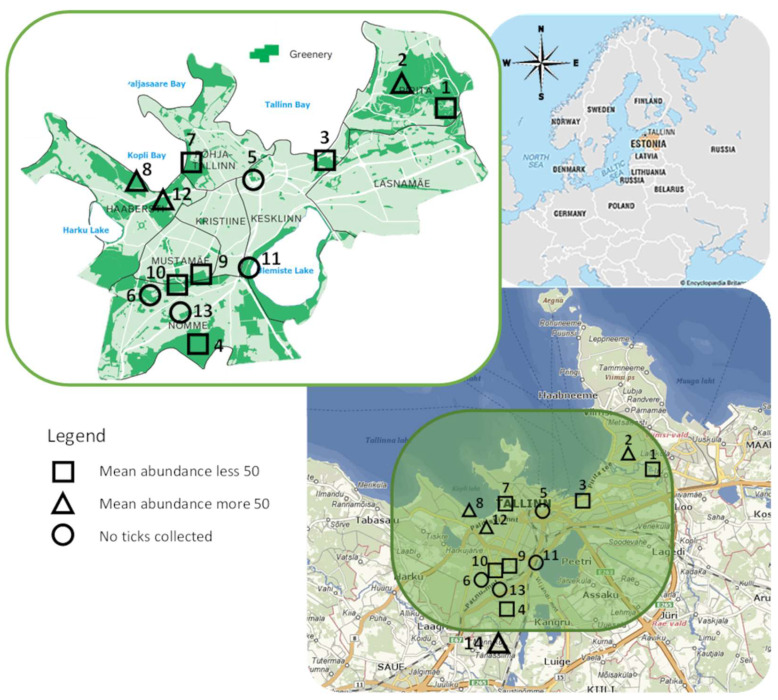
Tick collection sites, located in green areas within the city (Tallinn green area map https://statistika.tallinn.ee/, accessed 21 January 2021). Circular marks represent places with no ticks collected, triangular marked sites represent places with over 50 ticks collected and square marks represent places with 1 to 49 ticks collected. Site names according to numbers are as follows: 1—Pirita river valley, 2—Pirita forested park, 3—Kadrioru park, 4—Ilmarise health trails, 5—Hirve and Toompark, 6—von Glehni park, 7—Stroomi, 8—Estonian Open Air Museum, 9—Sütiste park, 10—Nõmme-Mustamäe, 11—Järve health trails, 12—Tallinn’s Zoo, 13—Sanatooriumi park, 14—Männiku.

**Table 1 microorganisms-12-01918-t001:** Tick collection sites and tick collection results.

Site No.	Name	Description	Latitude/Longitude	m^2^ Flagged	Total Ticks (M/F/N) *	IA	DOT (95%CI) DOA/DON **	Larvae ***
1	Pirita river valley	riverside with rich herbal lower vegetation and bushes	59.4574, 24.9023	450	18 (1/1/16)	6.0	4.0 (2.5–6.2); 0.4/3.6	−
2	Pirita forested park	large urban mixed forest, with hills and swamp areas and rich litter	59.4604, 24.8593	1250	122 (7/8/107);	40.7	9.8 (8.1–11.4); 1.2/8.6	+
3	Kadrioru	large urban park with mainly broadleaved trees	59.4415, 24.7982	300	6 (1/1/4);	2.0	2.0 (0.9–4.2); 0.7/1.3	−
4	Ilmarise health trails	large natural-like urban mixed forest with swamps	59.3659, 24.6666	1400	37 (5/2/30);	12.3	2.6/(1.8–3.5); 0.5/2.1	+
5	Hirve/Toompark	central city park, some bushes with a litter	59.4336, 24.7374	600	0 (0/0/0); −/−		−	−
6	von Glehni park	a park in the large urban mixed forest	59.3925, 24.6577	300	0 (0/0/0); −/−		−	−
7	Stroomi	urban broadleaved natural-like forest at the seaside	59.4372, 24.6921	1200	38 (6/5/27);	12.7	3.2 (2.3–4.3); 0.9/2.3	+
8	Estonian Open Air Museum	broadleaved- and mixed-type urban semi-forested area at the seaside	59.4323, 24.6395	1200	226 (18/33/175);	75.3	18.8 (16.7–21.1); 4.3/14.6	++
9	Sütiste forested park	urban mixed-type forest	59.3944, 24.6899	800	31 (3/3/25);	15.5	3.9 (2.5–5.2); 0.8/3.1	−
10	Nõmme-Mustamäe	urban mixed type forested area	59.38952, 24.6745	600	30 (3/1/26);	15.0	5.0 (3.5–7.0); 0.7/4.3	−
11	Järve health trails	semi-forested area, mainly with pine trees and an herbal lower layer	59.3997, 24.7299	600	0 (0/0/0); −/−		−	−
12	Tallinn Zoo	natural-like broadleaved forested areas with a rich lower layer	59.4208, 24.6616	1500	234 (30/39/165);	78.0	15.6 (13.9–17.5); 4.6/11.0	+++
13	Sanatooriumi park	semi-forested area, mainly with pine trees and an herbal or mossy lower layer	59.3762, 24.6638	600	0 (0/0/0); −/−		−	−
14	Männiku	mixed and coniferous forest with a mainly herbal or mossy lower layer	59.3273, 24.6797	400	73 (5/5/63);	36.5	18.3 (14.8–22.3); 2.5/15.8	−
Overall				11,200	815 (79/98/638)			

* N—nymphs, M—male, F—female; ** DOT, DOA and DON—density of total ticks, adults and nymphs, respectively, collected per 100 m^2^; IA—index of abundance = no. of ticks/all minutes of collection by all collectors ×60 (i.e., one-hour reduction index); *** L—larvae (“−“—no larvae observed; “+”—single larvae on some transects; “++”—a portion of larvae on several transects; “+++”—many larvae on several transects); the presence of larvae has been noted without exact count.

**Table 2 microorganisms-12-01918-t002:** Tick-borne pathogens detected in urban questing ticks.

					Borrelia					Rickettsiales			TBEV %; CI, 95%
Site No. *	Site Name *	Total No. of Ticks	TBPs % (No.); CI, 95% ***	DIN TBP ^¥^	BBSL, % (No.); CI, 95%	BA, % (No.)	BG, % (No)	BV, % (No)	*Bmiy* % (No.); CI, 95%	*Rh* % (No.); CI, 95%	An. ph, % (No.); CI, 95%	*N. mik*, % (No.); CI, 95%	
1	Pirita river valley	18	27.8% (5); 12.5–50.9%	0.9	0.0% (0)	0.0% (0)	0.0% (0)	0.0% (0)	0.0% (0)	22.2% (4); 8.5–45.8%	0.0% (0)	0.0% (0)	5.6% (1); 1–25.8%
2	Pirita forested park	122	31.1% (38); 23.6–39.8%	2.7	11.5% (14); 6.7–18.3%	4.9% (6)	4.9% (6)	1.6% (2)	0.0% (0)	18.0% (22); 42.2–72.2%	1.6% (2); 0.4–5.8%	3.3% (4); 1.3–8.1%	0.0% (0);
3	Kadrioru	6	16.7% (1); 3.0–56.4%	0.3	16.7% (1); 3.0–56.4%	0.0% (0)	16.7% (1)	0.0% (0)	0.0% (0);	0.0% (0);	0.0% (0)	0.0% (0)	0.0% (0);
4	Ilmarise health trails	37	24.3% (9); 13.4–40.1%	0.4	8.1% (3); 2.8–21.3%	0.0% (0)	2.7% (1)	2.7% (1)	0.0% (0); 0.0–9.4%	16.2% (6); 7.6–31.1%	0.0% (0)	0.0% (0)	2.7% (1); 0.5–13.8%
7	Stroomi	38	18.4% (7); 9.2–33.4%	0.5	13.2% (5); 5.8–27.3%	10.5% (4)	0.0% (0)	2.6% (1)	0.0% (0); 0.0–9.2%	7.9% (3); 2.8–20.8%	0.0% (0)	0.0% (0)	0.0% (0);
8	Estonian Open Air Museum	226	43.8% (99); 37.5–50.3%	6.3	25.2% (57); 20.0–31.3%	24.8% (56);	0.4% (1)	0.0% (0)	4.4% (10); 2.4–8.0%	13.7% (31) 9.8–18.8%	0.5% (1); 0.0–2.5%	7.5% (17); 4.8–11.7%	0.4% (1) 0.1–2.5%
9	Sütiste forested park	31	19.4% (6); 9.2–36.3%	0.8	16.1% (5); 7.1–32.6%	9.7% (3)	0.0% (0)	6.5% (2)	0.0% (0); 0.0–11.0%	3.2% (1); 0.6–16.2%	0.0% (0)	0.0% (0)	0.0% (0);
10	Nõmme-Mustamäe	30	3.3% (1); 0.6–16.7%	0.2	3.3% (1); 0.6–16.7%	0.0% (0)	0.0% (0)	3.3% (1)	0.0% (0); 0.0–11.4%	0.0% (0); 0.0–11.4%	0.0% (0)	0.0% (0)	0.0% (0);
12	Tallinn Zoo	234	41.5%(97); 35.3–47.9%	5.3	22.2% (52); 17.4–28.0%	20.9% (49)	0.9% (2)	0.0% (0)	3.9% (9); 2.0–7.2%	15.0% (35); 11.0–20.1%	0.4% (1); 0.1%–2.4%	10.2% (24); 7.0–14.8%	0.0% (0);
14	Männiku	73	19.2% (14); 11.8–29.7%	2.8	5.5% (4); 1.8–13.3%	4.1% (3)	0.0% (0)	0.0% (0)	2.7% (2); 0.8–9.5%	11.0% (8); 5.7–20.2%	0.0% (0)	0.0% (0)	1.4% (1); 0.2–7.4%
TOTAL		815	34.0% (277); 30.8–37.3%	2.4	17.4% (142); 15.0–20.2%	14.8% (121); 85.2% †	1.3% (11); 7.7% †	0.9% (7); 4.9% †	2.6% (21); 1.7–3.9%	13.5% (110); 11.3–16.0%	0.5% (4); 0.2–1.3%	5.5% (45); 4.2–7.3%	0.5%(4); 0.2–1.3%

* Sites with no ticks collected were excluded; *** TBPs—tick-borne pathogens; BBSL—*B. burgdorferi* s.l., BA—*B. afzelii*, BG—*B. garinii*, BV—*B. valaisiana*, Bmiy—*B. miyamotoi*; Rh—*R. helvetica*; An. ph—*A. phagocytophilum*, N. mik—*N. mikurensis*, TBEV—tick-borne encephalitis virus; *** CI—confidence interval, calculated only for sites with TBPs observed; ¥ DIN—density of infected nymphs per 100 m^2^, all tick-borne pathogens included; †—% from BBSL-positive 3.5. Rickettsia species.

## Data Availability

All additional data associated with this study can be obtained from the corresponding author on reasonable request. Unique nucleotide sequences obtained during this study were submitted to the GenBank database under the following accession numbers: MW924118—MW924135 for BBSL species, MW924974—MW924983 for *B. miyamotoi*, MW924984—MW925040 for *R. helvetica*, MW922757—MW922793 for *N. mikurensis*, MW922752—MW922756 for *A. phagocytophilum* and MW916612—MW916613 for TBEV.

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
