# Peer review of "Ticks and Tick-Borne Pathogens in Popular Recreational Areas in Tallinn, Estonia: The Underestimated Risk of Tick-Borne Diseases"

_microorganisms, 2024, doi:10.3390/microorganisms12091918_

Round 1

Reviewer 1 Report

Comments and Suggestions for Authors

Ticks and tick-borne pathogens in popular recreational areas in Tallinn, Estonia: an underestimated risk of tick-borne diseases,the manuscript has good reference value for urban tick-borne disease prevalence ,but there are some shortcomings. 

1. Some of the results are not clearly and succinctly described, e.g. 3.3.

2. Table 1,2 requires refined specifications.

3. The conclusions need to be refined and rewritten.

4.Check numbers and statistical analysis.

5. The discussion section should be strengthened and refined.

Comments on the Quality of English Language

/

Author Response

We want to extend our heartfelt thanks for careful review of our manuscript, " Ticks and tick-borne pathogens in popular recreational areas in Tallinn, Estonia: an underestimated risk of tick-borne diseases”.  We truly appreciate your insightful comments and constructive suggestions, which have been invaluable in guiding us to enhance the quality of our study.

Reviewer Comment 1: Some of the results are not clearly and succinctly described, e.g., 3.3.

We sincerely appreciate your insightful comment regarding the clarity and conciseness of the results section. We have thoroughly reviewed and revised this section, particularly subsection 3.3, to ensure the results are presented more clearly and succinctly. The revised text now provides a more straightforward and concise description of the findings, making it easier for readers to understand the key points. L210-235

Reviewer Comment 2: Table 1, 2 requires refined specifications.

Thank you for your valuable suggestion to refine the specifications in Tables 1 and 2. We have carefully reviewed both tables and made necessary adjustments to clarify the data presentation and improve readability. The specifications and captions have been refined to provide more precise and comprehensive information.

Reviewer Comment 3: The conclusions need to be refined and rewritten.

We appreciate your feedback on the conclusion section. In response, we have refined and rewritten the conclusions to better reflect the findings of our study and their implications. The revised conclusions now provide a more concise summary of our work and highlight the significance of our results in the context of the field. L445-468

Reviewer Comment 4: Check numbers and statistical analysis.

Thank you for pointing out the need to check the numbers and statistical analysis. We have meticulously reviewed all numerical data and statistical analyses in the manuscript. Corrections have been made where necessary to ensure accuracy and consistency throughout the text.

Reviewer Comment 5: The discussion section should be strengthened and refined.

We are grateful for your suggestion to strengthen and refine the discussion section. We have thoroughly revised this section to provide a more comprehensive analysis of our results in relation to existing literature. The revised discussion now offers deeper insights and addresses potential implications more effectively, thereby enhancing the overall quality of the manuscript. L312-443

Reviewer 2 Report

Comments and Suggestions for Authors

This paper describes the findings of significant tick-borne pathogens in urban recreational areas in Tallinn. The chapters on Materials and Methods and Results are clearly written. The Discussion is concise and well-organized, with a logical flow that comments on and compares the obtained results with those of other authors.

Minor comments:

L49-50. Understandable parts. What is meant by the expression “reaching oppositely record incidence”?

L50-53. It is necessary to divide this complex sentence into shorter sentences to improve clarity and readability.

L56-61. This sentence is also unnecessary complex. Please, use the data from the six papers you have cited and write a few clear sentences.

L62-65. This part of the text feels more like a conclusion. It's not badly written, but it would be more appropriate to write the aims of this research.

L75. Authors must specify that, based on the listed parameters, 14 sites were selected for tick sampling. As stated, "according to observations," is not correct. Also, if I am not mistaken, collaboration with local authorities is another parameter that was used to determine the sampling sites.

L91. „surveyed“ should be replaced with „examined“.

L219. Species should be written in italics.

L308-309. Be specific here. Write the “urban recreational areas in Tallinn, Estonia”.

L309-312. You should emphasize the result, stating that of all the collected ticks from the genus Ixodes, you identified the species Ixodes ricinus, except for one tick which belongs to the species Ixodes persulcatus.

Author Response

We would like to express sincere gratitude for taking the time to review our manuscript titled "Ticks and tick-borne pathogens in popular recreational areas in Tallinn, Estonia: an underestimated risk of tick-borne diseases." We greatly appreciate your thoughtful comments and valuable suggestions, which have been instrumental in helping us improve the quality of our work.

L49-50. Understandable parts. What is meant by the expression “reaching oppositely record incidence”?

Thank you for pointing this out. We are agree with this comment. The sentence has been revised for better clarity and logic. Estonia has long been an endemic area for TBE and LB, and by 2020, both diseases had reached record incidence rates.

L50-53. It is necessary to divide this complex sentence into shorter sentences to improve clarity and readability.

Thank you for pointing this out. We are agree with this comment. The sentence has been revised for better clarity and logic. The incidence of TBE fell nearly threefold, reaching its lowest rate in the last 20 years at 5.1 cases per 100,000 population. In contrast, the number of LB cases reached a record high during the same period, with 182.1 cases per 100,000 population. This is three times higher compared to the levels recorded in 2013.

L56-61. This sentence is also unnecessary complex. Please, use the data from the six papers you have cited and write a few clear sentences.

Thank you for pointing this out. In this case, the references are to articles that describe the first detection of specific pathogens in Estonia.

L62-65. This part of the text feels more like a conclusion. It's not badly written, but it would be more appropriate to write the aims of this research.

Thank you for pointing this out. We are agree with this comment. The sentence has been revised for better clarity and logic. The aim of this study is to investigate the distribution and abundance of ticks in urban areas of Tallinn and to analyze these ticks for the presence of tick-borne patho-gens (TBPs), including TBEV, BBSL, B. miyamotoi, A. phagocytophilum, N. mikurensis, and Rickettsia spp., in popular recreational and leisure sites within the city.

L75. Authors must specify that, based on the listed parameters, 14 sites were selected for tick sampling. As stated, "according to observations," is not correct. Also, if I am not mistaken, collaboration with local authorities is another parameter that was used to determine the sampling sites.

Thank you for pointing this out. The conditions for selecting the sites were described in more detail above. The sentence has been revised for greater clarity and meaning. Based on these criteria, 14 sites were selected in collaboration with the respective authorities where applicable.

L91. „surveyed“ should be replaced with „examined“.

Thank you for pointing this out. We are agree with this comment. The word „surveyed“ was replaced with „examined“. Each site was examined once.

L219. Species should be written in italics.

Thank you for pointing this out. We are agree with this comment. Sequence analysis of the 5S-23S IGS revealed the presence of three BBSL genospecies. The most dominant was B. afzelii (121/142; 85,2%), followed by B. garinii (11/142; 7,3%) and B. valaisiana (7/142; 4,7%), while three samples remained unspecified at the genospecies level due to poor sequencing results which in turn might imply on mixed genospecies.

L308-309. Be specific here. Write the “urban recreational areas in Tallinn, Estonia”.

Thank you for pointing this out. We are agree with this comment. Prior to the present study, there was no previous information on the spread of ticks and TBPs in urban recreational areas in Tallinn, Estonia.

L309-312. You should emphasize the result, stating that of all the collected ticks from the genus Ixodes, you identified the species Ixodes ricinus, except for one tick which belongs to the species Ixodes persulcatus.

Thank you for pointing this out. We are agree with this comment. This study confirms the presence of Ixodes tick species in popular recreational, outdoor sports, and leisure areas in the largest city of Estonia, with abundance rates that are comparable to or even exceed those previously recorded in the most endemic foci in natural environments. Notably, among all collected ticks from the genus Ixodes, we identified the species Ixodes ricinus, except for a single tick that belonged to the species Ixodes persulcatus. This finding underscores the diversity of tick species present in urban settings and highlights the potential public health risks associated with these areas.

Round 2

Reviewer 1 Report

Comments and Suggestions for Authors

The author has made a lot of revisions to the review comments, and it is suggested that they be published after minor revisions.

1.Statistical reconciliation and analysis.

2. Full sulk.

Comments on the Quality of English Language

/

Author Response

Dear Reviewer,

Comment 1.

We sincerely appreciate your review of our manuscript and the opportunity to improve it. To generate the statistical data, we used the article by Kazimírová M et al., 2023 (https://doi.org/10.3390/microorganisms11071666) as an example. In our case, we were unable to calculate the dependence of pathogen presence in ticks on factors such as air temperature, number of visitors, month of the year, or the abundance of small animals. In our study, the ticks were collected only once at each of the sampling sites.

In our article, we calculated Questing Tick Abundance as the density of the total number of ticks (DOT), adults (DOA), and nymphs (DON) collected per 100 m². Furthermore, confidence intervals with 95% (CI) for TBP prevalence and the Chi-square test were used to evaluate the significance of results for each study site. These data are presented in Tables 1 and 2, as well as in the Results section (lines 206, 208, 216-217).

Comment 2. Full sulk.

Unfortunately, the authors did not fully understand this comment and are unable to make the corresponding changes to the text.

Thank you once again for your valuable feedback.